# Rabies Virus-Neutralizing Antibodies in Free-Ranging Invasive Wild Boars (*Sus scrofa*) from Brazil

**DOI:** 10.3390/pathogens13040303

**Published:** 2024-04-07

**Authors:** Patricia Parreira Perin, Talita Turmina, Carmen Andrea Arias-Pacheco, Jonathan Silvestre Gomes, Lívia de Oliveira Andrade, Natália de Oliveira Zolla, Talita Oliveira Mendonça, Wilson Junior Oliveira, Willian de Oliveira Fahl, Karin Correa Scheffer, Rene dos Santos Cunha Neto, Maria Eduarda Rodrigues Chierato, Enio Mori, Artur Luiz de Almeida Felicio, Guilherme Shin Iwamoto Haga, Maria Carolina Guido, Luiz Henrique Barrochelo, Affonso dos Santos Marcos, Estevam Guilherme Lux Hoppe

**Affiliations:** 1Department of Pathology, Reproduction and One Health, São Paulo State University, Jaboticabal 14884900, Brazil; patricia.perin@unesp.br (P.P.P.); talita.turmina@unesp.br (T.T.); carmen.arias@unesp.br (C.A.A.-P.); jonathan.silvestre@unesp.br (J.S.G.); livia.andrade@unesp.br (L.d.O.A.); n.zolla@unesp.br (N.d.O.Z.); talita.mendonca@unesp.br (T.O.M.); junior.oliveira@unesp.br (W.J.O.); 2Laboratory of Rabies Diagnosis, Pasteur Institute, São Paulo 01311090, Brazil; wofahl@pasteur.saude.sp.gov.br (W.d.O.F.); ksferreira@pasteur.saude.sp.gov.br (K.C.S.); renecunhaneto@yahoo.com.br (R.d.S.C.N.); mchierato1@gmail.com (M.E.R.C.); enio@usp.br (E.M.); 3Agricultural Defense Coordination, Department of Agriculture and Supply of the State of São Paulo, Campinas 13070178, Brazil; artur.felicio@unesp.br (A.L.d.A.F.); guilherme.haga@sp.gov.br (G.S.I.H.); maria.guido@sp.gov.br (M.C.G.); lbarrochelo@sp.gov.br (L.H.B.); amarcos@sp.gov.br (A.d.S.M.)

**Keywords:** invasive alien species, hunting, Suidae, *Rhabdoviridae*, zoonosis

## Abstract

Rabies, one of the most lethal global zoonoses, affects all mammals. It remains circulating worldwide in sylvatic cycles through terrestrial and airborne reservoirs, and in Brazil, bats are currently the main reservoirs and source of transmission. Wild boars, an important invasive alien species in Brazil, are a proven food source for hematophagous bats and may participate in the Brazilian sylvatic cycle of rabies. We evaluated the presence of this pathogen in hunted wild boars from the São Paulo state using histopathology, the direct fluorescent antibody test (DFA), viral isolation in cell culture (VICC), the rapid fluorescent focus inhibition test (RFFIT), and quantitative reverse transcription polymerase chain reaction (RT-qPCR). The results of histopathological, DFA, VICC, and RT-qPCR analysis were negative for all samples; seven serum samples tested positive in the RFFIT, and titers ranged from 0.13 IU/mL to 0.5 IU/mL. The presence of rabies virus-neutralizing antibodies in the studied wild boars suggests the circulation of the virus in these animals. Educative actions directed at hunters should include information on the prevention of this important zoonosis.

## 1. Introduction

Rabies, one of the most lethal global zoonoses, is an under-reported, poorly funded, and under-diagnosed re-emerging neglected tropical disease (NTD) in the majority of low- and middle-income nations [1,2]. It affects all mammals, and it is caused by neurotropic viruses from the genus *Lyssavirus*, family *Rhabdoviridae*, and order *Mononegavirales* [3]. The main reservoirs of rabies viruses (RABV) used to be dogs, but following the implementation of mass vaccination, this pathogen remains circulating worldwide in Latin America in sylvatic cycles through terrestrial and airborne reservoirs [4]. Nevertheless, dogs remain a significant source of human rabies cases in many parts of the world, particularly in Asia and Africa [5]. In Brazil, vaccination campaigns have significantly decreased canine rabies, and bats are currently the main source of transmission [2]. The relevance of wild animals as potential persistent reservoirs and the sources of reintroduction and spillover of RABV are still largely unknown and should be a priority for epidemiological investigations of rabies on a landscape scale [6].

Wild boars (*Sus scrofa*) are an important invasive alien species in Brazil due to their deleterious environmental and socioeconomic impact, and currently the main population control strategy in the country is hunting [7]. They are a proven food source for hematophagous bats and could act as rabies sentinels [8,9,10]. Rabies infections have been reported in a wild boar following three attacks on people in Kerala, India [11]. Two studies, one from Heilongjiang province, China, and one from Goiás and Paraná states, Brazil, found RABV-neutralizing antibodies in free-ranging wild boars from areas free from wildlife rabies vaccination [10,12]. Teider-Júnior et al. [10] suggested that invasive wild boars may participate in the Brazilian sylvatic rabies cycle.

Considering the lack of epidemiological studies on RABV exposure in free-ranging invasive wild boars in Brazil and the possible risk to human and animal health, we evaluated the presence of this pathogen in hunted wild boars from the São Paulo state using histopathology, the direct fluorescent antibody test (DFA), viral isolation in cell culture (VICC), the rapid fluorescent focus inhibition test (RFFIT), and quantitative reverse transcription polymerase chain reaction (RT-qPCR).

## 2. Materials and Methods

### 2.1. Study Area and Sample Acquisition

The study was conducted in 11 cities within São Paulo state’s northeastern region and covered an area of approximately 6470 km^2^ (Figure 1). The vegetation cover is a transition between the Cerrado savanna and Atlantic rainforest Biomes [13], and the study area is strongly anthropized, with few forest remnants. The predominant climate is humid subtropical [14]. For the reference period of 1991 to 2020, the average annual temperatures were between 20 °C and 22 °C, and the annual precipitation accumulation was between 1400 and 1600 mm [15]. The regional economy is strongly dependent on agricultural activities, primarily sugarcane, soy, citrus, corn, coffee, tomato, and peanut cultivation, as well as beef cattle and poultry farming [16].

Between 2018 and 2022, our team accompanied registered wildlife controllers for wild boar hunting, and samples from 82 animals were obtained. The controllers adopted two hunting strategies: at night, they preferred active search with spotlights and long-distance firearms; during the daytime, they used hunting dogs and short-distance firearms. The sampling procedure did not adhere to biostatistical criteria as it relied on the hunting success of the partner hunters.

Immediately after death, fragments of the central nervous system (CNS) of 75 wild boars were collected. Part of the fragments were placed in isothermal boxes with ice for transportation, while others were preserved in a 10% buffered formaldehyde solution at room temperature. Additionally, 72 whole blood samples were collected via cardiac puncture using disposable sterile syringes and vacuum collection tubes without anticoagulants and placed in isothermal boxes with ice for transport. All the samples were then sent to the Parasitic Diseases Laboratory (LabEPar) of São Paulo State University, Jaboticabal, Brazil. The CNS fragments in bags, along with serum samples obtained by centrifugation of the blood samples, were stored in freezers at −20 °C before being dispatched to the virology sector of the Rabies Diagnosis Laboratory at the Pasteur Institute in São Paulo, Brazil, in isothermal boxes with ice, for further analysis. The remaining CNS fragments were kept in the 10% buffered formaldehyde solution at room temperature for 24 to 48 h until they underwent further processing at the Parasitic Diseases Laboratory of São Paulo State University, Jaboticabal, Brazil.

### 2.2. Bat Roost Identification

The bats’ roosts were registered based on an on-site search for suspicious locations that may contain active colonies of the hematophagous bat *Desmodus rotundus*. The methodology considers the bats’ behavior and common habits, access to prey (mostly production animals, such as ruminants, horses, and pigs), access to water, and protection from predators. The coordinates (Sirgas 2000) were obtained at the entrance of the roost.

### 2.3. Histopathology

After fixation, 48 CNS fragments were embedded in paraffin and then cut into 3 μm thick sections that were adhered to glass slides and stained with hematoxylin-eosin for histopathological analysis. The glass slides were examined under an optical microscope Olympus BX-51 (Olympus America Inc., Center Valley, PA, USA) at various magnification levels to identify possible alterations.

### 2.4. Direct Fluorescent Antibody Test

The DFA assay was conducted in duplicate on 46 non-autolyzed CNS impression smears, following the protocol described by Dean et al. [17]. Readings were performed by fluorescence microscopy (A1, AX10; Carl Zeiss, Blauvelt, NY, USA).

### 2.5. Viral Isolation

The 46 non-autolyzed CNS fragments were macerated and diluted in 40 mL fetal bovine serum (Gibco, Grand Island, NY, USA), 2 mL antibiotic and antifungal (Sigma Aldrich, Burlington, MA, USA), 17 g NaCl (Merck, Darmstadt, Germany), and distilled water q.s.p. 2000 mL in the proportion of 20% sample and 80% diluent. The diluted macerate was centrifuged at 1500× *g* at 4 °C for 30 min (Heraeus Megafuge 40R Centrifuge^®^; Thermo Scientific, Waltham, MA, USA). The supernatant was stored at −20 °C until viral isolation. VICC was performed in duplicate with murine neuroblastoma cell strain (N2A) in 96-well flat-bottomed microplates (Corning, Charlotte, NC, USA), according to Webster and Casey [18] and modified by Castilho et al. [19]. The reaction was read on a fluorescence microscope (A1, AX10; Carl Zeiss, Thornwood, NY, USA) with an HBO-100 lamp at 200× magnification.

### 2.6. Detection of RABV Virus-Neutralizing Antibodies

The RFFIT was conducted in duplicate on 72 serum samples, following the method described by Smith et al. [20], with some modifications [21]. A human international standard rabies immune globulin (SRIG) reference serum of 30 IU/mL (National Institute for Biological Standards and Control, Herts, UK) recognized by the World Health Organization (WHO) was diluted to 0.5 IU/mL. The SRIG and the test serum samples were then diluted in six serial fivefold dilutions starting from 1:2.5 in 96-well flat-bottomed microplates (Corning, Charlotte, NC, USA) in a final volume of 50 µL using Eagle’s minimum essential medium (Corning, Charlotte, NC, USA) with Earle’s balanced salt solution (Merck, Darmstadt, Germany) supplemented with 10% fetal bovine serum (Gibco Laboratories, Grand Island, NY, USA) (MEM-FBS). A volume of 50 µL of a challenge virus standard RABV strain suspension (CVS-132-11A, Pasteur Institute, São Paulo, Brazil) containing 100 FFD_50_ was added to the diluted serum, and the plates were incubated at 37 °C for 90 min at 5% CO_2_. Then, 100 µL of BHK-21 cells suspension (2.5 × 10^5^ cells/mL) was added, and the plates were incubated again at 37 °C for 20 h at 5% CO_2_. The cells were fixed with 80% acetone at −20 °C. The reaction was revealed with the addition of rabies antivirus conjugate produced by the Pasteur Institute [22]. The reading was performed on an inverted fluorescence microscope at 200× magnification, and 18 fields were examined in each plate well to verify the presence of fluorescent foci. The titers were calculated by comparison to standard serum using the Spearman–Kärber method of analysis. Samples were considered positive when the results were equal to or greater than 0.10 IU/mL [23].

### 2.7. Quantitative Reverse Transcription Polymerase Chain Reaction

RNA extraction was performed on 24 autolyzed CNS samples using the RNeasy Lipid Tissue Mini extraction kit (Qiagen, Hilden, Germany) following the manufacturer’s instructions. The extracted RNA was subjected to a Pan-Lyssavirus Taqman RT-qPCR Assay, as described [24], detailed, and updated [25] previously. Primers and probes used in the RT-qPCR assay can be found at https://doi.org/10.1371/journal.pone.0197074.t001, accessed on 6 April 2024. Briefly, the assay involved a single-step RT-qPCR reaction that amplified the lyssavirus RNA genome (LN34), along with a single-step RT-qPCR reaction that amplified the host β-actin mRNA as an internal negative control.

For the RT-qPCR reaction, a standardized protocol was followed using the AgPath-ID™ One-Step RT-PCR Kit (Life Technologies, Carlsbad, CA, USA). Each reaction contained 2.0 µL of template RNA, 1.0 μL of forward and reverse primer sets (at a concentration of 10 μM each), and 1.0 μL of probe (at a concentration of 5 μM), which were combined in a final reaction volume of 25 μL according to the manufacturer’s instructions. To ensure accuracy, each sample was tested in duplicate, and a positive control (CVS-132-11A, Pasteur Institute, São Paulo, Brazil) and a negative control (MiliQ™ ultrapure water) were included in each assay. The reactions were run on MicroAmp™ optical 96-well plates using a 7500 Real-Time PCR System (Applied Biosystems, Bedford, MA, USA). The cycling conditions for the RT-qPCR reaction were initial reverse transcription at 50 °C for 30 min, followed by inactivation of reverse transcriptase and initial denaturation at 95 °C for 10 min, 45 cycles of amplification at 95 °C for 15 s, and a final extension at 56 °C for 30 s. The samples were classified based on the number of quantification cycles (Cq) into positive (Cq < 35), inconclusive (Cq between 35 and 40), or negative (Cq > 40).

### 2.8. Statistical Analyses

The frequency rate was estimated as the number of positive samples in the DFA, VI, RIFFIT, and rt-qPCR tests divided by the total of samples (72 just for the RIFFIT test and 79 overall) and expressed as a percentage (%), and 95% confidence intervals (CIs) were calculated using the Wilson method. All statistical analyses were performed using the software R version 4.2.1 and the significance level was set to 0.05.

## 3. Results

The results of DFA, VICC, and RT-qPCR analysis were negative for all samples. However, seven serum samples obtained from wild boars hunted in Monte Azul (four samples), Paraíso (one sample), Cajobi (one sample), and Colina (one sample) cities tested positive in the RFFIT (Figure 2). The overall estimated frequency of RABV infection in wild boars was 8.86% (7/79; 95% CI 4.36–17.18%), as shown in Table 1. Considering only the RFFIT, as this was the only test with positive results, the overall seroprevalence was 9.72% (7/72; 95% CI 4.47–18.74%), as shown in Table 2. The RFFIT-positive animals’ titers ranged from 0.13 IU/mL to 0.5 IU/mL (0.13 IU/mL, *n* = 1; 0.14 IU/mL, *n* = 3; 0.17 IU/mL, *n* = 1; 0.27 IU/mL, *n* = 1; 0.5 IU/mL, *n* = 1).

Due to the non-random sampling method and the limited sample size of this study, it was not possible to determine if there was a significant association between age, geographical mesoregion, and frequency.

There were no histopathological alterations in the one CNS sample that corresponded with the serum of an animal that tested positive in the RFFIT. The most common histopathological alterations were perivascular cuffing (4/48; 8.33%), neuronophagia (13/48; 27.08%), vacuolization (8/48; 16.66%), edema (8/48; 16.66%), congestion (2/48; 4.16%), and hemorrhage (3/48; 6.25%). No Negri body was detected in any sample. A summary of results obtained for wild boar samples in RIFFIT, DFA, viral isolation, rt-qPCR, and histopathological techniques can be found in Appendix A.

## 4. Discussion

The overall prevalence of rabies observed in this study, 8.86% considering all executed tests and 9.72% in RFFIT only, is slightly lower than the 11% observed in a previous study in Brazilian wild boars [10]. Rabies virus-neutralizing antibodies (RVNA) were detected in wild animals in Brazil, suggesting exposure [26] and even active infection [27].

Wild boars are generalist, opportunist omnivore mammals that feed on the carcasses of dead animals occasionally [28,29]. As the RABV virus can be transmitted through cannibalism and scavenging in natural environments [30], this could be one alternative to explain the contact of wild boars with this pathogen. However, it is noteworthy to mention that wild boars only use this resource in cases of food scarcity [28,31], and food is available all year in the study area due to agricultural activities.

On the other hand, attacks by vampire bats *Desmodus rotundus* (Phyllostomidae: Desmodontinae) are common and were described in two Brazilian Biomes [8,10,32]. These bats are endemic to the Neotropical region and are considered the main reservoirs of RABV, being responsible for outbreaks in livestock [33,34]. *Desmodus rotundus* usually roosts in caves and tree hollows, but they can adapt to human constructions such as abandoned houses, corrals, and other buildings, resulting in the presence of this bat species both in conserved and anthropized areas. In contrast to other bat species that are negatively affected by deforestation, *D. rotundus* is more common in pasture areas as they feed on livestock [35]. Although our data did not reveal associations between bat roosts and RVNA in the wild boars, even considering the flying range of these animals, not all bat roosts in the most affected area may be known.

The presence of RVNA in titers ranging from 0.1 to 0.5 IU/mL suggests viral circulation among the animals, possibly due to a non-lethal or abortive infection. Despite controversies on the relevance of RVNA’s presence, they are frequently detected in bats, unvaccinated wildlife, and domestic dogs in endemic areas [26,27,36]. False positives could be the result of cytotoxic samples due to contamination during collection, as frequently noted in field research [37]. In this study, all the samples were collected by the research team by cardiac puncture, reducing the risk of sample contamination. At the laboratory, hemolyzed and/or turbid samples were discarded for serological testing.

Two probes were used in the RT-qPCR assay, one for RABV and one for endogenous β-actin mRNA. No amplification in the β-actin assay may indicate extraction failure, poor sample quality, PCR inhibition, or RNA degradation [25], but we considered the result as negative only when there was amplification with β-actin with Ct < 33, which occurred for all the samples tested.

The higher frequency of RVNAs in females could be related to their gregarious behavior. Wild boars’ sounders are matriarchal groups led by dominant females, with submissive females, juveniles, and piglets, while the adult males tend to live alone [38]. These groups with several individuals could be more attractive to *D. rotundus* vampire bats, as they tend to feed in more densely populated areas [35]. A randomized study with a larger sample size should be conducted to verify if there is a relationship between sex and the frequency of RVNAs in wild boars. In Brazil, domestic dogs were historically associated with RABV reservoirs and main vectors [39]. However, from 2010 to 2023, most cases of human rabies in Brazil were related to contact with wild animals, including twenty-four cases associated with bats, five with non-human primates, and two with foxes [4]. Additionally, there were four documented incidents of human rabies originating from cats infected with a bat variant through secondary transmission. The most recent case of human rabies in Brazil caused by a canine-origin variant occurred almost 9 years ago, following a dog epizootic in the region near an endemic country like Bolivia [4]. In São Paulo state, the Pasteur Institute registered 1.467 cases of animal rabies from 2018 to 2022. Most cases, 658, are related to cattle, followed by non-hematophagous bats, with 570 cases. The virus was detected in only 25 *Desmodus rotundus* vampire bats, although these animals are the main vectors of rabies in cattle [40,41]. The rabies vigilance in Brazil depends on the submission of samples of suspect animals to the Pasteur Institute, so the cases do not represent the real prevalence of the disease. The wild boars included in this study were hunted in pastures used for cattle rearing or in forest remnants and riparian forests adjacent to these pastures, so the transmission could be related to the same bat roost. Unfortunately, there is no data on the presence of RABV in the detected bat roosts.

The efforts of the Brazilian Ministry of Health and other stakeholders engaged in rabies control reached the goal of zero dog-mediated rabies cases in 2008. After that, the importance of wildlife as reservoirs and vectors of rabies increased. In contrast to other parts of the world, where oral baiting is used to deliver the RABV vaccine to wild animals [42], there is no vaccine-mediated prophylaxis program directed at wildlife in Brazil [43]. Human and/or domestic animals’ prophylaxis is recommended after rabies detection in wild animals [43]. In this case, the detection of RVNA in the wild boars is not related to oral vaccines as reported in other countries [44,45,46].

All the wild boars demonstrated normal behavior before hunting, as most of them were hunted while feeding, while others started fleeing as soon as they perceived human presence. Also, none of the animals presented skin lesions, which are suggestive of bat feeding or a recent animal attack. Domestic pigs infected by hematophagous bats exhibit signs of paralytic rabies, with pelvic limb paralysis [47]. Wild boars presenting this clinical manifestation possibly would be hidden in denser parts of the riparian forests or forest remnants, hindering their capture by hunters using a spotlight, but the hunting dogs could easily find them. All the wild boars captured by dogs presented the expected behavior when cornered. They had no suggestive clinical signs, and on the histopathological examination of the CNS samples, no Negri bodies or specific alterations could be observed. The dogs used in the captures are vaccinated annually with cell culture vaccines and were observed daily by the hunters.

Since 2013, when the Brazilian environmental authority authorized lethal control of wild boars by registered hunters, interest in this activity and wild boars’ meat consumption have increased gradually. For comparison, in 2019, there were 44,258 hunters and 150 hunting enterprises registered for wild boar population control. This number increased considerably in 2022, with 136,528 hunters and 361 companies registered [48]. In consequence, hunting activity represents a higher risk for the transmission of zoonotic diseases because of increased direct contact with blood and fluids from killed animals [49]. In São Paulo state alone, the second Brazilian state in the number of registered hunters [48], 84% of hunters report that they regularly consume wild boar meat [50]. Serology results from a study [9] indicated a lack of effective immune protection against rabies in hunters from two Brazilian states, Paraná and Goiás. They also found that only 18% (9/49) of hunters reported rabies vaccination, and 67% (6/9) had their titers previously tested. Although undocumented, the transmission of RABV from swine to humans is feasible, although low [51], as these animals are susceptible to RABV infection [52,53,54]. Considering carcass manipulation, the human health risk posed by wild boars is unknown, but it is well documented that the manipulation of dog carcasses may transmit rabies to humans [55,56,57,58].

The state of São Paulo implemented actions for wild boars’ pathogen vigilance and carcass destination, aiming to reduce the risk to human and animal health [59]. Educative actions directed at the hunters are promoted periodically to present information on diseases, including zoonoses, related to wild boars, including information on sample collection and carcass disposal. Considering the possible risk of RABV transmission to this group, this zoonosis should be emphasized in the course contents, contributing to rabies prevention in humans and hunting dogs.

## 5. Conclusions

The presence of RVNAs in the studied wild boars suggests the circulation of the virus in these animals. Educative actions directed at hunters should highlight information on the prevention of this important zoonosis.

## Figures and Tables

**Figure 1 pathogens-13-00303-f001:**
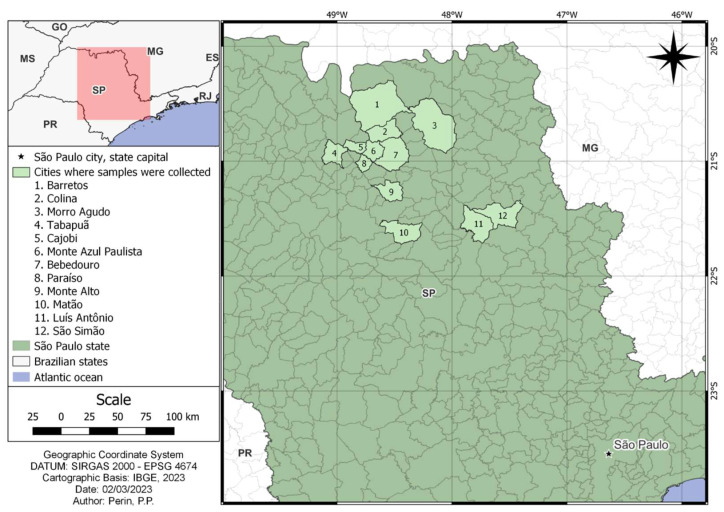
Map highlighting the cities where samples from invasive free-ranging wild boars (*Sus scrofa*) were collected, São Paulo state, Brazil. The area highlighted in red corresponds to the larger map’s area.

**Figure 2 pathogens-13-00303-f002:**
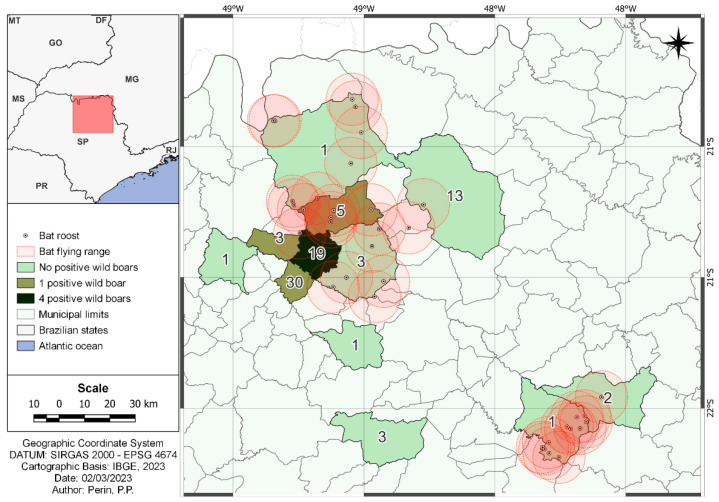
Map highlighting the number of serum samples that tested positive in the RFFIT test, along with the location of bat roosts within a 10 km perimeter of their flying range. The numerical values within each city’s area represent the number of wild boars sampled at each location.

**Table 1 pathogens-13-00303-t001:** Overall prevalence of rabies, considering DFA, VICC, RT-qPCR, and RIFFIT, in free-ranging wild boars (*n* = 79) from São Paulo state, Brazil.

Variable	Rabies % (p/*n*, CI)
Sex	
Male	2.3% (1/43, 0.4–12.1%)
Female	16.7% (6/36, 7.9–31.9%)
Age	
≤6 months	5.0% (1/20, CI = 0.9–23.6%)
>6 months	10.2% (6/59, CI = 4.7–20.5%)
Mesoregion (cities)	
Araraquara (Matão)	0% (0/3, CI = 0–56.1%)
São José do Rio Preto (Cajobi, Paraíso, Tabapuã)	5.9% (2/34, CI = 1.6–19.1%)
Ribeirão Preto (Barretos, Bebedouro, Colina, Luís Antônio, Monte Alto, Monte Azul Paulista, Morro Agudo, São Simão)	11.9% (5/42, CI = 5.2–25.0%)

**Table 2 pathogens-13-00303-t002:** Seroprevalence (RFFIT) of rabies in free-ranging wild boars (*n* = 72) from São Paulo state, Brazil.

Variable	Rabies % (p/*n*, CI)
Sex	
Male	2.6% (1/38, 0.5–13.5%)
Female	17.6% (6/34, 8.3–33.5%)
Age	
≤6 months	5.6% (1/18, CI = 1.0–25.7%)
>6 months	11.1% (6/54, CI = 5.2–22.2%)
Mesoregion (cities)	
Araraquara (Matão)	0% (0/2, CI = 0–65.7%)
São José do Rio Preto (Cajobi, Paraíso, Tabapuã)	6.1% (2/33, CI = 1.7–19.6%)
Ribeirão Preto (Barretos, Bebedouro, Colina, Luís Antônio, Monte Alto, Monte Azul Paulista, Morro Agudo, São Simão)	13.5% (5/37, CI = 5.9–28.0%)

## Data Availability

The original contributions presented in the study are included in the article/Appendix A, further inquiries can be directed to the corresponding author.

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
