# Peer review of "Rabies Virus-Neutralizing Antibodies in Free-Ranging Invasive Wild Boars (Sus scrofa) from Brazil"

_pathogens, 2024, doi:10.3390/pathogens13040303_

Round 1

Reviewer 1 Report

Comments and Suggestions for Authors

The paper explores exposure of alien invasive species sus scrofa wild boar in Brazil to Rabies virus and attempts to suggest they are a risk to ongoing sylvatic source rabies now that dog rabies is under good control.  It is a good paper with sound methodology but in short it is overinterpreted and the absolute facts are cryptic. I would like a more transparent and less biased view in the narrative and caution in the conclusions. The dataset is too small and unrandomised to get a good handle on prevalence and since sylvatic cases in humans are also now very low ascribing risk very difficult. I have no problem with speculation on the role of boar potentially but this must be made clear. Weaknesses in the statistical results are not clearly stated and must be to help the reader to move beyond the abstract  narratives. I have made comments on the PDF.

Author Response

Response to Reviewer 1

Thank you very much for taking the time to review this manuscript. Please find the detailed responses below and the corresponding revisions/corrections highlighted in yellow in the re-submitted files.

Point-by-point response to Comments and Suggestions for Authors:

Comment 1: The paper explores exposure of alien invasive species sus scrofa wild boar in Brazil to Rabies virus and attempts to suggest they are a risk to ongoing sylvatic source rabies now that dog rabies is under good control.  It is a good paper with sound methodology but in short it is overinterpreted and the absolute facts are cryptic. I would like a more transparent and less biased view in the narrative and caution in the conclusions. The dataset is too small and unrandomised to get a good handle on prevalence and since sylvatic cases in humans are also now very low ascribing risk very difficult. I have no problem with speculation on the role of boar potentially but this must be made clear. Weaknesses in the statistical results are not clearly stated and must be to help the reader to move beyond the abstract narratives. I have made comments on the PDF.

Response 1: Thanks for your careful review and observations. We will consider them.

Comment 2: Line 40The main reservoirs of rabies viruses (RABV) used to be dogs, but following the implementation of mass vaccination, this pathogen remains circulating worldwide in sylvatic cycles through terrestrial and airborne reservoirs[4].”
The reference (4) is not looking at this question, if you have a reference to justify this statement please enter it. I am afraid globally the majority of human rabies cases are still contracted from domestic dogs so I don't believe this is true. Yes in some countries such as the US the incidence now of about 2-3 human rabies cases annually is from one or two wildlife species. This extremely small number is good but also should be noted to give proportionality to the sylvatic rabies risk. Here the use of vaccine treatment for exposure has also reduced cases significantly. 

Response 2: Thank you for your observation. We included the clarification “In Latin America” to the text and changed the reference supporting the information to Schneider et al., 2023 (https://doi.org/10.3390/pathogens12111342). We also included the statement: “Nevertheless, dogs remain a significant source of human rabies cases in many parts of the world, particularly in Asia and Africa [5].”

[5]: Hampson et al., 2015 (https://doi.org/10.1371/journal.pntd.0003709). 

Comment 3: Line 44 The relevance of wild animals as potential persistent reservoirs sources of reintroduction and spillover of RABV is still largely unknown and should be a priority for epidemiological investigations of rabies on a landscape scale [5].”

Although understanding sylvatic cycles and risks is important it is equally important not to not overstate the risk or to demonise wildlife. Lyssaviruses are a large group and are not going away whilst we have biodiversity and this fact should be recognised. Whilst elimination of the evolved canine rabies pathogen is a reasonable aim. In Europe sylvatic rabies cases are extremely rare and usually occupational with bat exposure for example - about 2 cases a decade.

Response 3: We respectfully disagree, as the reference we are citing explicitly states that “the role of wildlife remains widely unknown in terms of a potential constant reservoir, as a source of reintroduction or spillover, thus it remains neglected in most of the rabies control programs”, and “Here, we conclude that the control of rabies at the domestic-wild animal interface on a landscape-scale is vital, and the inclusion of wild animals in rabies control programs is crucial.” Regarding wildlife as subjects of surveillance, as the passage suggests, is not to demonize.

Comment 4: Line 47Wild boars (Sus scrofa) are an important invasive species in Brazil due to its deleterious environmental and socioeconomic impact and currently the main population control strategy in the country is hunting [6].”
Consider using the term alien invasive species (IUCN Definition: Invasive alien are animals, plants or other organisms that are introduced by humans, either intentionally or accidentally, into places outside of their natural range, negatively impacting native biodiversity, ecosystem services or human economy and well-being. This categorises them appropriately.

Response 4: Done.

Comment 5: Line 54Teider-Júnior et al. [9] suggested that invasive wild boars may participate in the Brazilian sylvatic rabies cycle.”

This paper made no attempt to quantify this risk which may be very small and the occupational risk of hunters was highlighted but the fact there was negativity serologically in a co-sample in this study in hunters suggests if there is a risk currently it is negligible?

Response 5: We respectfully disagree, as the risks are unknown; therefore, they cannot be considered insignificant.

Comment 6: Line 59The rapid fluorescent focus inhibition test (RFFIT) (…)”

You should mention that this test has been validated for use in human medicine and has been validated for animal testing and used in pigs to confirm infection post exposure - references available.

Response 6: The RFFIT procedure, which has been standardized and validated, is considered the gold standard for detecting rabies antibodies in both animals and humans. A rabies antibody level exceeding or equal to 0.5 IU/mL, as per WHO and WOAH guidelines, signifies an appropriate response to rabies vaccination in humans and animals or human antemortem diagnosis of rabies in serum or CSF samples. Additionally, the Pasteur Institute holds accreditation for this technique in international interlaboratory proficiency testing by ANSES, serving as the European Union Reference Laboratory for Rabies, particularly for animal samples.

https://food.ec.europa.eu/animals/movement-pets/approved-rabies-serology-laboratories/non-eu-countries_en#brasil

Comment 7: Line 87 “The CNS fragments in bags, along with serum samples obtained by centrifugation of the blood samples, were stored in freezers at -20°C before being dispatched to the virology sector of the Rabies Diagnosis Laboratory at the Pasteur Institute in São Paulo”

Rabies virus preservation by freezing at -20 is not suitable for long term storage unless cryopreservant is used. Some information on delays to processing samples to support methodology or use of cryopreservants such as sucrose 60% should be mentioned.

https://doi.org/10.1590/0037-8682-0135-2013

Response 7: While we agree that the preservation method was not ideal and may pose a limitation, it's important to note that at the Pasteur Institute, one probe was specifically used in the qPCR assay as a means of sample quality control method. All samples were considered adequate, as stated in our text: “Two probes were used in the RT-qPCR assay, one for RABV and one for endogenous 230 β-actin mRNA. No amplification in the β-actin assay may indicate extraction failure, poor sample quality, PCR inhibition, or RNA degradation [24], but we considered the result as negative only when there was amplification with β-actin with Ct < 33, which occurred for all the samples tested.”

Comment 8: Line 1642.8. Statistical analyses”

Since this is not a random sample this result is not true prevalence and this fact should be stated it is an estimate only and yes you state it is estimated but it would be good to be more explicit on what this means.

Response 8: We corrected it to “frequency”.

Comment 9: Line 175 “The overall prevalence of RABV infection in wild boars was 8.86% (7/79; 95% CI 4.36% - 17.18%)”

Estimate.

Response 9: We corrected it to “estimated frequency”.

Comment 10: Line 177Considering only the RFFIT, as this was the only test with positive results, the overall seroprevalence was 9.72% (7/72; 95% CI 4.47% - 18.74%).”

At this level with a non random sampling method may mean very little or mask an even smaller exposure rate almost within the error confidence for the test and method of sampling used. I dont see this critique in the discussion.

Response 10: We included the phrase: “Due to the non-random sampling method and the limited sample size of this study, it was not possible to determine if there was a significant association between age, geographical mesoregion, and frequency” in the Results section to address this concern.

Comment 11: Line 185When considering all diagnostic techniques (RFFIT, DFA, viral isolation, and RT-qPCR) as shown in Table 2, females were 8.2 times more likely to become infected.”

This is a stretch in imagination...sample too small a result too small to be confident in this result meaning that much. The odds ratios are also odd. Why would males be less at risk? I think this result is spurious potentially.

Response 11: We removed “The Fisher’s exact test was used to evaluate the association between sex, age, geographical mesoregion, and prevalence” from our methodology, and modified the passage “There was no significant association between age, geographical mesoregion, and prevalence. When considering all diagnostic techniques (RFFIT, DFA, viral isolation, and RT-qPCR) as shown in Table 2, females were 8.2 times more likely to become infected. When considering only the RFFIT, as shown in Table 3, females were 7.73 times more likely to become infected.“ from the Results section, to: “Due to the non-random sampling method and the limited sample size of this study, it was not possible to determine if there was a significant association between age, geographical mesoregion, and frequency”. We also removed the “Fisher’s Exact Test P-value” column from Tables 2 and 3.

Comment 12: Line 201 “The overall prevalence of rabies observed in this study, 8.86% considering all executed tests and 9.72% in RFFIT only, is slightly lower than the 11% observed in a previous study in Brazilian wild boars [9].”

See Dascula et al 2019 which shows a rabies prevalence in wild boar of 42.31% in a much larger sample size. I think this suggests the situation in Brazil is very different and exposure patterns probably reflecting the bat issue. Please broaden the discussion a bit to put this into context and with more of a risk or proportional perspective.

Response 12: The study mentioned does not show a rabies prevalence in wild boar of 42.31%; instead, 42.31% were found to have rabies antibodies after oral rabies vaccination (ORV) campaigns. This study does not reflect Brazil’s epidemiological situation, as Brazil is more biodiverse and doesn’t employ ORV campaigns. It also does not mention bats’ relevance to rabies sylvatic cycles. Therefore, we respectfully disagree that adding it to our paper would broaden the discussion.

Comment 13: Line 206 “As the RABV virus can be transmitted through cannibalism and scavenging in natural environments [29], this could be one alternative to explain the contact of wild boars with this pathogen. However, it is noteworthy to mention that wild boars only use this resource in cases of food scarcity [27, 30], and food is available all year in the study area due to agricultural activities.”

If this were the case I would expect similar results comparing Europe to Brazil.

Response 13: Again, we respectfully disagree. In this aspect, Brazil’s epidemiological situation, as mentioned above, is also diverse. Unlike Europe, where food availability is highly influenced by winter, Brazil does not experience seasonality in food availability: “winter is a critical season for the wild boar, along with other European mammals, as the natural food supply can be considerably limited by climatic conditions.“ Mikulka et al., 2018, https://doi.org/10.25225/fozo.v67.i3-4.a3.2018.

Comment 14: Line 235 “The higher frequency of RVNAs in females, 7.73 or 8.2 times more prone to infection  than males, depending on the tests considered in the analyses, could be related to their gregarious behavior. Wild boars’ sounders are matriarchal groups led by dominant females, with submissive females, juveniles, and piglets; while the adult males tend to live alone [37].”

I think you need to be cautious with this result.

Response 14: We removed the passage “7.73 or 8.2 times more prone to infection than males, depending on the tests considered in the analyses” from the Discussion to address this issue, and added: “A randomized study with a larger sample size should be conducted to verify if there is a relationship between sex and the frequency of RVNAs in wild boars.”. We believe, however, that we are being cautious by only suggesting there might be a relationship and explaining why that could be in that paragraph.

Comment 15: Line 241 “In Brazil, domestic dogs were historically associated as RABV reservoirs and main vectors [38]. However, most cases of human rabies in Brazil, from 2010 to 2022, are related to contact with wild animals, including bats [39].”

This is clearly an important result to report but it is also important to state what the absolute case numbers are and that ~20% cases remain dog associated so the actual number of sylvatic origin cases is even smaller overall. Not dissimilar now to the northern regions of the Americas.

Response 15: We changed the text to: “In Brazil, domestic dogs were historically associated as RABV reservoirs and main vectors [38]. However, from 2010 to 2023, most cases of human rabies in Brazil were related to contact with wild animals, including twenty-four cases associated with bats, five with non-human primates, and two with foxes [4]. Additionally, there were four documented incidents of human rabies originating from cats infected with a bat variant through secondary transmission. The most recent case of human rabies in Brazil caused by a canine-origin variant occurred almost 9 years ago, following a dog epizootic in the region near an endemic country like Bolivia [4]” to address this concern.

Comment 16: Line 255 “In contrast to other parts of the world, where oral baiting is used to deliver RABV vaccine to wild animals [42], there is no vaccine-mediated prophylaxis program directed to wildlife in Brazil [43].”

This is unrealistic in a biodiverse country like Brazil. The oral baiting in the biodiversity poor European region is not a good case to compare costs and benefits or efficacies of this approach.

Response 16: We agree that ORV for wildlife is unrealistic in a biodiverse country like Brazil. However, in the passage mentioned, we did not imply that ORV is feasible in Brazil; we merely highlighted that ORV is not utilized in Brazilian wildlife.

Comment 17: Line 283 “They also found that only 18% (9/49) of hunters reported rabies vaccination and 67% (6/9) had their titers previously tested. Although undocumented, the transmission of RABV from swine to humans is feasible [51], as these animals are susceptible to RABV infection [52-54].”

But it would be better to put a risk parameter on this...I would say probably very low or negligible from the evidence presented here. Not zero of course.

Response 17: We included “although low” in the passage to address this.

Comment 18: Line 297 “The presence of RVNAs in the studied wild boars suggests the circulation of the virus in these animals. Educative actions directed to hunters should highlight information on the prevention of this important zoonosis.”

I dont share this conclusion as I am not convinced by the data. I would prefer more caution and more data is needed.

Response 18: We would like to keep this passage as it is because we are suggesting it rather than affirming it. We addressed concerns about caution previously by removing some statistical data that didn’t seem trustworthy, as explained in the comments above.

Reviewer 2 Report

Comments and Suggestions for Authors

Maybe also ELISA test should use to detect rabies antibodies.. 

Author Response

Response to Reviewer 2                 

Thank you very much for taking the time to review this manuscript. Please find the detailed responses below.

Point-by-point response to Comments and Suggestions for Authors:

Comment 1: Maybe also ELISA test should use to detect rabies antibodies..

Response 2: Thank you for your suggestion.
The RFFIT procedure, which has been standardized and validated, is considered the gold standard for detecting rabies antibodies in both animals and humans. Additionally, the Pasteur Institute holds accreditation for this technique in international interlaboratory proficiency testing by ANSES, serving as the European Union Reference Laboratory for Rabies, particularly for animal samples.

https://food.ec.europa.eu/animals/movement-pets/approved-rabies-serology-laboratories/non-eu-countries_en#brasil

Brazilian government organizations, responsible entities for product approval, such as the Ministry of Agriculture, Livestock, and Food Supply (MAPA), as well as the Health Regulatory Agency (ANVISA), do not currently possess any Rabies Virus Antibodies ELISA Kits approved for diagnostic purposes in animals or humans. ELISA may be discouraged due to the possibility of producing false positive outcomes by detecting antibodies, especially considering that the ELISA cutoff is not standardized.

Reviewer 3 Report

Comments and Suggestions for Authors

Dear authors, 
The title of the study intrigued me and was very promising. When reading the article however, I encountered some things that puzzled me and should be clarified: 

line 40: the main reservoir of rabies viruses used to be dogs, but following the implementation of mass vaccination, this pathogen remains circulating worldwide in sylvatic cycles through terrestrial and airborne reservoirs: Dogs are still a huge reservoir for the virus and the main drivers for infection in humans worldwide. In addition, the reference indicated does not state that dogs are no longer the main reservoir, mainly because mass vaccination is still a huge challenge, mainly in developing countries. If this applies to the situation in Brazil, please specify that this is the situation in Brazil.

From the article it is unclear how many animals were included into the study. In line 80 it is mentioned that fragments of the nervous systems of 70 wild boars were collected and additionally 72 blood samples were collected. Is this from the same animals? Later on (line 106) it is mentioned that only 46 samples were used for DFA and virus isolation and 24 (line 142) were used for PCR. Why were not all samples tested for both techniques? 
The article would benefit from a summerizing table stating which samples were tested with which technique. 

In line 176 the prevalence of rabies infection was calculated as 7 out of 79 animals. But it is nowhere mentioned that 79 animals were sampled. Where does this number come from?

Also it is confusing to speak of "rabies infection" when diagnostic techniques did not show any infection or presence of the virus in these animals. In the discussion you refer to citation 26 stating active infection, whereas in this publication it is mentioned that the nature of the exposure cannot be determined and that the titers suggest contact with the virus. 

Author Response

Response to Reviewer 3

Thank you very much for taking the time to review this manuscript. Please find the detailed responses below and the corresponding revisions/corrections highlighted in blue in the re-submitted files.

Point-by-point response to Comments and Suggestions for Authors:

Comment 1: Dear authors, 
The title of the study intrigued me and was very promising. When reading the article however, I encountered some things that puzzled me and should be clarified: 

line 40: the main reservoir of rabies viruses used to be dogs, but following the implementation of mass vaccination, this pathogen remains circulating worldwide in sylvatic cycles through terrestrial and airborne reservoirs: Dogs are still a huge reservoir for the virus and the main drivers for infection in humans worldwide. In addition, the reference indicated does not state that dogs are no longer the main reservoir, mainly because mass vaccination is still a huge challenge, mainly in developing countries. If this applies to the situation in Brazil, please specify that this is the situation in Brazil.

Response 1: Thank you for your observation. We included the clarification “In Latin America” to the text and changed the reference supporting the information to Schneider et al., 2023 (https://doi.org/10.3390/pathogens12111342). We also included the statement: “Nevertheless, dogs still remain a significant source of human rabies cases in many parts of the world, particularly in Asia and Africa [5].”

[5]: Hampson et al., 2015 (https://doi.org/10.1371/journal.pntd.0003709). 

Comment 2: From the article it is unclear how many animals were included into the study. In line 80 it is mentioned that fragments of the nervous systems of 70 wild boars were collected and additionally 72 blood samples were collected. Is this from the same animals? Later on (line 106) it is mentioned that only 46 samples were used for DFA and virus isolation and 24 (line 142) were used for PCR. Why were not all samples tested for both techniques? The article would benefit from a summerizing table stating which samples were tested with which technique. In line 176 the prevalence of rabies infection was calculated as 7 out of 79 animals. But it is nowhere mentioned that 79 animals were sampled. Where does this number come from?

Response 2: We created a table summarizing the samples tested for each test (Table S1). Not all samples underwent all techniques due to constraints in fieldwork. Serum or CNS samples couldn't always be collected from the same animals and rt-qPCR was used on samples considered not suitable for DFA and VI (Pasteur Institute's internal work protocol). In addition to the table, we provided supplementary information to clarify the sample distribution.
Line 74: Between 2018 and 2022, our team accompanied registered wildlife controllers for wild boar hunting, and samples from 82 animals were obtained.
Line 166: Frequency rate was estimated as the number of positive samples in the DFA, VI, RIFFIT, and rt-qPCR tests divided by the total of samples (72 just for the RIFFIT test and 79 overall), and expressed as a percentage (%), and 95% confidence intervals (CI) were calculated using the Wilson method. (To make it clearer which results were used to obtain the frequency and confidence intervals).

Comment 3: Also it is confusing to speak of "rabies infection" when diagnostic techniques did not show any infection or presence of the virus in these animals. In the discussion you refer to citation 26 stating active infection, whereas in this publication it is mentioned that the nature of the exposure cannot be determined and that the titers suggest contact with the virus.

Response 3: Done.

Round 2

Reviewer 3 Report

Comments and Suggestions for Authors

NA